# GPR55 Receptor Activation by the *N*-Acyl Dopamine Family Lipids Induces Apoptosis in Cancer Cells via the Nitric Oxide Synthase (nNOS) Over-Stimulation

**DOI:** 10.3390/ijms22020622

**Published:** 2021-01-09

**Authors:** Mikhail G. Akimov, Alina M. Gamisonia, Polina V. Dudina, Natalia M. Gretskaya, Anastasia A. Gaydaryova, Andrey S. Kuznetsov, Galina N. Zinchenko, Vladimir V. Bezuglov

**Affiliations:** Shemyakin-Ovchinnikov Institute of Bioorganic Chemistry, Russian Academy of Sciences, Ul. Miklukho-Maklaya, 16/10, Moscow 117997, Russia; alyafonya@mail.ru (A.M.G.); polinadudkinz@gmail.com (P.V.D.); natalia.gretskaya@gmail.com (N.M.G.); a.bosaya@oxylipin.ibch.ru (A.A.G.); andrej.kuznecov@phystech.edu (A.S.K.); zgn55@yandex.ru (G.N.Z.); vvbez@ibch.ru (V.V.B.)

**Keywords:** *N*-acyldopamines, GPR55, *N*-acyldopamines cytotoxicity, nitric oxide synthase, cancer cell apoptosis, endocannabinoids

## Abstract

GPR55 is a GPCR of the non-CB1/CB2 cannabinoid receptor family, which is activated by lysophosphatidylinositol (LPI) and stimulates the proliferation of cancer cells. Anandamide, a bioactive lipid endocannabinoid, acts as a biased agonist of GPR55 and induces cancer cell death, but is unstable and psychoactive. We hypothesized that other endocannabinoids and structurally similar compounds, which are more hydrolytically stable, could also induce cancer cell death via GPR55 activation. We chemically synthesized and tested a set of fatty acid amides and esters for cell death induction via GPR55 activation. The most active compounds appeared to be *N*-acyl dopamines, especially *N*-docosahexaenoyl dopamine (DHA-DA). Using a panel of cancer cell lines and a set of receptor and intracellular signal transduction machinery inhibitors together with cell viability, Ca^2+^, NO, ROS (reactive oxygen species) and gene expression measurement, we showed for the first time that for these compounds, the mechanism of cell death induction differed from that published for anandamide and included neuronal nitric oxide synthase (nNOS) overstimulation with concomitant oxidative stress induction. The combination of DHA-DA with LPI, which normally stimulates cancer proliferation and is increased in cancer setting, had an increased cytotoxicity for the cancer cells indicating a therapeutic potential.

## 1. Introduction

GPR55 is a G-protein coupled receptor (GPCR) of the non-CB1/CB2 cannabinoid receptor family [1]. GPR55 is linked to Gα_12_, Gα_13_, and Gα_q/11_ subunits [2]. The natural ligand of GPR55 is thought to be lysophasphatidylinositol (LPI); it usually activates the receptor in the concentration range of 0.5–4 µM. In addition, other bioactive lipids of the endocannabinoid family like 2-arachidonoylglycerol and anandamide (AA-EA) are able to activate GPR55 or serve as its inverse agonists [3].

GPR55 is widely expressed in various cancers and is usually associated with the late stages and bad prognosis of the disease [2]. The molecular mechanism of this association is the induction of cell proliferation after receptor activation by its natural agonist LPI, which is abundantly synthesized and released by cancer cells [4]. The LPI signal is transduced via Ras homolog family member A (RhoA) or Ca^2+^ to extracellular signal-regulated kinase (ERK) and nuclear factor of activated T-cells (NFAT), thus leading to an increase of cell proliferation [1].

GPR55 is known to be linked to diverse downstream signaling pathways. Depending on the ligand and cell type, the receptor could be coupled to Gα_13_, Gα_q/11_, or Gα_12_ subunits with subsequent activation of one of RhoA, phosphatidylinositol-3-kinase–phospholipase Cγ, Rho-associated protein kinase (ROCK), or Rac proteins, followed by Ca^2+^ liberation or ERK and NFAT activation (cf. review [1]).

In view of the pro-proliferative activity of GPR55 in cancer, its inhibitors are viewed as potential antitumor agents. Thus, several GPR55-selective antagonists were synthesized [5] and demonstrated to inhibit cancer growth [6] and reduce cancer chemoresistance [6]. However, in some cancers, e.g., cholangiocarcinoma, GPR55 activation has an antiproliferative effect, and thus the clinical use of the antagonists of this receptor is controversial [2]. At the same time, GPR55 activation by the endogenous cannabinoid AA-EA was shown to induce cell death in osteosarcoma [7], and such activity is quite promising. However, AA-EA as a possible therapeutic molecule has two major drawbacks: it is quickly hydrolyzed by the ubiquitously present fatty acid amide hydrolase (FAAH) and similar to some GPR55 antagonists, could be a psychoactive substance [8]. In addition, in some cancers AA-EA and other cannabinoids are pro-proliferative [9], which could be due to difference in the GPR55 downstream signaling, induced by AA-EA and LPI [10,11,12]. As such, AA-EA was able to increase intracellular Ca^2+^ concentration, but without oscillation, characteristic for LPI [11], and failed to induce ERK activation and β-arrestin binding of the receptor [10,12].

Besides AA-EA, other fatty acid amides with dopamine [13,14,15], serotonin [16] and ethanolamine [17] are described. These substances may either have their own receptor targets or share them with AA-EA [18,19] and are synthesized and metabolized by the organism [20]. At the same time, they may lack psychic activity and be much more hydrolytically stable compared to this molecule [16,21]. Moreover, *N*-acyldopamines (NADA) [22,23] and other unsaturated fatty acid amides [24] are even known to elicit cell death in various cancer cell lines. For example, in MCF-7, SKBR3 and MDA-MB-231 breast cancer cells, dopamine amides of docosahexaenoic (DHA-DA) and eicosapentaenoic (EPA-DA) acids were shown to induce autophagy and cell death via Peroxisome proliferator-activated receptor γ (PPARγ) receptor activation [25]. A similar activity was observed for the ethanolamides of docosahexaenoic (DHEA) and eicosapentaenoic (EPEA) acids on the MCF-7 breast cancer cells [26]. On the prostate cancer cell lines LNCaP and PC3, these substances were antiproliferative as well. For EPEA, but not DHEA, this effect was in part mediated by CB1 and CB2 receptor activation [27]. Interestingly, the oxidized forms of EPEA and DHEA have higher antiproliferative potential than their precursors [28].

We hypothesized that other endocannabinoids and structurally similar compounds, which are not substrates for the FAAH, could also induce cancer cell death via GPR55 activation and have no stimulatory effect on cancer cells.

In this paper, we tested a set of fatty acid amides and esters for cell death induction via GPR55 activation. The most active compounds appeared to be NADA, especially DHA-DA. We showed for the first time that the mechanism of cell death induction by these compounds differed from that of anandamide and included neuronal nitric oxide synthase (nNOS) overstimulation with concomitant oxidative stress induction.

## 2. Results

### 2.1. Chemical Synthesis of Fatty Acid Esters

For this work, we synthesized several groups of substances (Figure 1).

First, we synthesized anandamide (AA-EA) as the known GPR55 endocannabinoid ligand. Then, we synthesized several AA-EA analogues with potential enhanced stability: amides of arachidonic acid with 2-chloroethylamine (AA-CEA) (analog of ethanolamine in which the hydroxyl group is replaced by chlorine atom), and cyclopropylamine (AA-CPA). Finally, we extended the substance set with virodamine (VRD) and dopamine amides of arachidonic (AA-DA), oleic (Ol-DA), and docosahexaenoic (DHA-DA) acids, and arachidonoyl serotonin (AA-5HT) based on our prior data on their cytotoxicity [23]. The substances were synthesized using the standard methods with yields of about 70% and purity of about 95−99% (microcolumn HPLC analysis, ESI-MS and ^1^H NMR).

### 2.2. Selection of the Most Active Fatty Acid Derivative Family in Cancer Cell Death Induction

The first step of the study was to check whether the synthesized compounds were able to induce cell death. To this end, we used the rat PC12 cell line as the model (without the differentiation induction), as we had previously observed a GPR55-linked apoptosis induction by NADA in this cell line [29]. After a 24-h incubation, all the compounds tested displayed cytotoxicity (Figure 2) with EC_50_ in range 6–80 µM (Table 1), while anandamide demonstrated the lowest activity. Dopamine amides of various fatty acids were the most active ones, and thus we chose one of them DHA-DA as the compound with the lowest EC_50_ for further studies.

### 2.3. DHA-DA, LPI and AA-EA Docking to the GPR55 Molecule

As far as DHA-DA behaved as a biased agonist for the GPR55 receptor in biological activity tests, we performed a molecular docking study to compare its interaction with the binding site of the receptor with two known GPR55 ligands, AA-EA and LPI. The structure of GPR55 in the active state from the work [5] was used. First, we docked each molecule to the whole receptor as one large binding area and clustered and averaged the coordinates of the docking results to locate potential alternative binding sites. Then the docking was repeated to a narrow zone around the new binding sites.

For DHA-DA, in total five binding sites (nearby GPR55 amino acid residues: site 2, Tyr106, Phe110, Leu148; site 3, Phe102, Pro155, Ile156; sites 1, 4, and 5, Phe182, Val163, Phe159) were detected (Figure 3). LPI displayed a single binding site, which overlapped with one of the DHA-DA sites. AA-EA possessed a major binding cluster, which partially overlapped with LPI and DHA-DA one, and a minor binding cluster, which was unique for this molecule.

The affinity of all three molecules to the DHA-DA sites estimated using molecular docking showed that by decreasing the binding efficiency the sites could be arranged in the following sequences for each molecule: LPI 3 > 1 > 4 > 5 > 2, AA-EA 1~3 > 4~5 > 2, DHA-DA 3 > 1 > 5~4 > 2 (Table 2), and the binding score of DHA-DA for the top LPI cluster was even lower (affinity higher), than for LPI itself.

### 2.4. GPR55 Knockdown Effect on DHA-DA Cytotoxicity

To validate the requirement of the GPR55 expression for DHA-DA cytotoxicity, we used siRNA knockdown to inactivate this receptor. We transfected the PC12 cells with a combination of three siRNAs, checked the appropriate mRNA decrease using RT-PCR and GPR55 protein decrease using Western blotting, and tested DHA-DA cytotoxicity on the transfected cells. The siRNA treatment indeed induced a ~80% decrease of the GPR55 signal in the cells (Figure 4A–D), and the DHA-DA cytotoxicity significantly diminished on the cells transfected with the GPR55 siRNA, but not on the cells transfected with a scrambled siRNA (Figure 4E). Therefore, GPR55 presence was indeed required for the DHA-DA cytotoxicity.

### 2.5. DHA-DA Cytotoxicity in the Cell Lines with the GPR55 Receptor

To validate the assumption that GPR55 participates in the DHA-DA cytotoxicity independently of the cell line, we tested the cytotoxicity of the substance on a panel of human cancer cell lines, chosen to represent different tissues: breast cancer MDA-MB-231, glioblastoma U-87 MG, pancreatic cancer PC3, PANC-1, and DU-145, colon cancer SW620, and a nontumorigenic epithelial cell line MCF-10A. The acute cytotoxicity was evaluated after a 24 h incubation with MTT assay detection. DHA-DA induced cell death in all cell lines tested with a general tendency of higher cytotoxicity in the cell lines with GPR55 receptor blocker activity (Figure 5A, Table 3). Given that MCF-10A could be considered as a noncancer control cell line, in all further experiments the DHA-DA concentration was chosen so as not to be toxic for this cell line (i.e., below 40 µM). The cytotoxicity was partially prevented by the GPR55 receptor blockers CID16020046 and cannabidiol (Figure 5B). These data indicated that GPR55 indeed played a significant role in DHA-DA cytotoxicity. However, the cytotoxicity inhibition by the receptor blockers was only partial, pointing to the participation of other receptors in DHA-DA cytotoxicity.

### 2.6. DHA-DA Interplay with LPI on Cancer Cells

LPI is a known endogenous agonist for the GPR55 receptor [4]. As far as DHA-DA behaved as a GPR55-biased agonist, we hypothesized that these two compounds should interfere with each other’s activity. To test this hypothesis, we treated the MDA-MB-231 cells with a combination of DHA-DA and LPI for 24 h and evaluated the viability of the cells using the MTT assay. LPI alone caused a statistically nonsignificant increase in cell viability (Figure 6A), while its combination with DHA-DA had an increased cytotoxicity (Figure 6B).

### 2.7. nNOS-Dependend ROS Production after the GPR55 Activation with DHA-DA during the Apoptosis Induction

The next question of the study was: what is the molecular mechanism of the apoptosis induction by GPR55 activation by DHA-DA? Earlier we observed that cell death induction by NADA could be blocked by GPR55 antagonists [29]. NADA also stimulated oxidative stress with NO and reactive oxygen species (ROS) generation in the process, and the removal of both ROS (reactive oxygen species) and NO prevents cytotoxicity [30,31]. We hypothesized that there could be a causative link between GPR55 activation and ROS and NO generation during the apoptosis induction by NADA. To test this hypothesis, we evaluated ROS and NO production after DHA-DA application with the blocker of GPR55; DHA-DA combinations with the inhibitors of NOS, nNOS (neuronal NO synthase), and ROS or NO scavengers were included as a control, and PC12 cells served as the model. DHA-DA treatment induced nNOS expression (Figure 7A–D), and the application of the GPR55 blocker did prevent both ROS and NO production (Figure 7). In this work, we used nondifferentiated PC12 cells as a model. According to the literature data, neither eNOS (endothelial NO synthase), nor iNOS (inducible NO synthase) are expressed in such cells [32,33]; they only appear after a treatment with the differentiation inducting stimuli [34]. As far as no mRNA expression of other NO synthase isoforms was detected, and the DHA-DA stimulated production of NO was completely prevented by the inhibitor of nNOS, no further analysis of other NOS isoforms was performed. Therefore, GPR55 activation by DHA-DA, indeed, leads to NOS activation and NO and ROS production.

### 2.8. Signal Transduction from GPR55 to nNOS

To evaluate the signal transduction from GPR55 to nNOS during cell death induction, we treated the PC12 cells with DHA-DA in combination with the inhibitors of various intracellular targets, chosen based on the literature data on GPR55 signaling (Figure 8).

To check the order of the subcellular target activation, several measurements were performed: cytotoxicity (expected to be decreased with the inhibitor of any cellular component, participating in cell death induction), NO generation (expected to be decreased for the components upstream of NOS inhibition), and ROS generation (expected to be decreased for the NOS and upstream components inhibition). In addition, we included the inhibitors of several ROS-sensitive kinases, participating in cell death induction (c-Jun N-terminal kinase (JNK), p38, and apoptosis-signal regulated kinase 1 (ASK1)) and mitochondrial permeability transition pore (MPTP) inhibitor, and directly measured the characteristic of inositol triphosphate receptor (IP3R) activation Ca^2+^ response and cAMP response element-binding protein (CREB) activation (Figure 9). Most blockers, except for KN-62 (calcium/calmodulin activated kinase II, CaMKII), prevented DHA-DA cytotoxicity, but only those upstream of NO generation prevented NO and ROS accumulation, which agreed with the hypothesis (Figure 5, Figure 9A–C). CREB activation was detected as expected (Figure 9D). Intracellular Ca^2+^ level was increased after the DHA-DA treatment. BAPTA and EGTA prevented this increase only partially, and GPR55 blocker O-1918 reduced the duration of the increased Ca^2+^ concentration presence in the cells (Figure 9E).

## 3. Discussion

In this study, we performed a search among the known and potential endocannabinoids and similar molecules for the substances that would be able to induce cancer cell death via the GPR55 receptor. Such activity is already described for anandamide, but this molecule is very unstable; thus, analogs with a similar activity and enhanced hydrolytic stability could be very useful. Among the tested substances, fatty acid amides with dopamine, especially DHA-DA, appeared to be most active. We demonstrated that this molecule indeed activates GPR55 in various cancer cell lines and induces cell death via this receptor. The mechanism of this activity in PC12 cells appeared to be NOS activation via increased Ca^2+^ signaling.

To perform the study, a panel of endocannabinoids and their analogs were synthesized using standard methods; the obtained yields and purity were typical for such substances. All of the synthesized compounds were cytotoxic for the GPR55-bearing PC12 cell line with EC_50_ in the range of 6–80 µM (Table 1). It should be noted that in this test anandamide and its ester analogue virodamine demonstrated the lowest cytotoxic activity. Dopamine amides of various fatty acids were the most active ones, and thus we chose one of them DHA-DA as the compound with the lowest EC_50_ for further studies. The observed cytotoxicity of DHA-DA was similar to that already described by us [23,31,35]. The cytotoxicity of several other anandamide analogs, on the other hand, was demonstrated for the first time. Overall, the compounds with a phenolic amide moiety displayed a higher cytotoxicity, which is in line with the data on the GPR55-linked cytotoxicity of 1,4-naphthoquinones for the MDA-MB-231 cells [36]. Considering a higher similarity of a phenolic ring (compared to the ethanolamine in AA-EA) to the inositol residue in the natural GPR55 agonist LPI, it could be hypothesized that the phenolic compounds have a higher affinity to this receptor.

Molecular docking studies further confirmed the possibility of the interaction of DHA-DA with the GPR55 LPI binding site. The molecule docking areas for LPI, DHA-DA, and AA-EA highly overlapped (Figure 3), and the calculated binding energies were very close (Table 2).

The cytotoxicity of DHA-DA was substantially decreased after the GPR55 siRNA knockdown (Figure 4), which is in line with our previous data on NADA cytotoxicity prevention by GPR55 antagonists and its pharmacological downregulation [29]. A scrambled siRNA control did not affect DHA-DA cytotoxicity, indicating that this indeed was a GPR55-based activity.

In all cell lines, except MCF-10A and U-87MG, DHA-DA cytotoxicity was at least partially blocked by both of the two GPR55 antagonists used, indicating that GPR55 is a cell line independent target for cell death induction by this compound (Figure 5). These data agree with the literature data on the GPR55 receptor expression in these cell lines [4,37,38,39,40,41].

In this experiment series, we used CBD (cannabidiol) and CID16020046 as the GPR55 blockers, both at the concentration of 400 nM. At higher concentrations, CBD acts as a weak CB1 receptor antagonist (IC_50_ 3.35 µM [42]), TRPV1 receptor agonist (EC_50_ 3.5 µM, [43]) and CB2 receptor inverse agonist (IC_50_ 27.5 µM [42]). CID16020046, on the other hand, is a GPR55 inverse agonist [44], which was demonstrated to block LPI-induced Ca^2+^ signaling (IC_50_ 0.21 μM in HEK-GPR55 cells), ERK1/2 phosphorylation and GPR55-mediated transcription factor activation [45]. In view of these data, the ability of both of these substances to block DHA-DA most probably indicates GPR55 involvement, and not their off-target effects.

For most of the tested cell lines, the cytotoxicity of DHA-DA was characterized for the first time. The fact that the GPR55 blockers prevented the cytotoxicity only partially could be explained by the participation of other cannabinoid and vanilloid receptors in this process. Such activity would be in line with the reported CB1-linked [46] and TRPV1-linked [47] cytotoxicity and with the ability of NADA to activate these receptors [48,49].

DHA-DA treatment of PC12 cells induced nNOS expression and NO and ROS accumulation, and the application of the GPR55 blocker prevented both ROS and NO production (Figure 7). These results are in line with the ability of NADA and other vanilloids to induce oxidative stress [31,50]. However, the data on the link between this event and GPR55 activation are novel. Based on the fact that no mRNA for other NOS isoforms was detected, and the inhibitor of nNOS fully prevented DHA-DA-induced NO accumulation, we concluded, that iNOS and eNOS did not participate in the DHA-DA action and did not study them further.

In the context of nNOS expression activation by DHA-DA, it should be noted, that several fatty amides, including (but not limited to) those binding to cannabinoid receptors, show anti-inflammatory or general immune modulating properties with DHEA being the most potent one [19,20,51]. The documented mechanisms include cyclooxygenase COX-2 inhibition by DHEA [52] and DHA-DA [53]. DHAE also reduces the inducible NO synthase iNOS and cytokine monocyte chemotactic protein-1 MCP-1 gene expression [54]. In addition, *N*-docosahexaenoyl serotonin attenuates IL-23-IL-17 signaling in macrophages [55]. Besides the anti-inflammatory activity, DHEA at nanomolar concentrations promotes neurogenesis, neurite outgrowth and synaptogenesis in developing neurons [51]. The seeming contradiction of observed NO generation stimulation by DHA-DA with these data is removed by the fact that nNOS is not a part of the inflammatory response, but rather is involved in neuroplasticity and blood flow regulation [56].

Both ROS and NO generation were also blocked with the NOS inhibitor L-NAME, and thus the first thought was to attribute all of the detected ROS to NO production. However, NO scavenger alone did not fully remove the ROS accumulation, and thus both NO and some other form of ROS were generated. Therefore, a more plausible explanation of these data is the ability of nNOS to synthesize H_2_O_2_ as a by-product as was shown previously [57]. The selective inhibitor of nNOS ARL 1477 completely prevented DHA-DA cytotoxicity and NO formation, and thus nNOS is the only NO synthase isoform participating in the observed activity. The produced H_2_O_2_ and NO may further react to form singlet oxygen [58], and thus a question arises whether this further interaction is the key player in the observed cytotoxicity effect. The fact that the addition of the NO scavenger PTIO reduces cytotoxicity only by 50%, indicates that NO-H_2_O_2_ interaction is indeed important, but H_2_O_2_ is enough for cytotoxicity to appear.

Based on the literature data on GPR55 signaling [1,59], we tested the influence of several downstream signal transduction components on the cytotoxicity, NO and ROS synthesis induced by DHA-DA. Indeed, DHA-DA induced CREB activation and Ca^2+^ accumulation in the cells. Cytotoxicity was reduced by all inhibitors tested, except for the CaMKII, but NO and ROS generation were not affected by the inhibition of supposed targets downstream of NOS. Based on these data and the data on NO, ROS, and cytotoxicity response to GPR55 blockers, we constructed a signal transduction scheme for cell death induction by DHA-DA (Figure 8). For the NADA and other endovanilloids this pathway is novel, but at the same time it does not introduce any novel components to the known GPR55 signal transduction, which could be expected for the observed activity switch from pro-proliferative for LPI to pro-apoptotic for DHA-DA. This seeming controversy could be explained by a much longer intracellular Ca^2+^ increase after DHA-DA treatment (Figure 9) compared to the literature data on LPI [60]. This difference may lead to a concomitant overstimulation of both nNOS expression and activity and ROS accumulation as a result and a shift from proliferation stimulation to cell death induction, which was already described in the literature for low and high concentration of this enzyme in cancer cells [61].

The combination of the natural GPR55 agonist LPI and DHA-DA quite surprisingly had an increased cytotoxicity, and this activity was observed at several DHA-DA concentrations and at the concentration of LPI which stimulated cell proliferation by itself (Figure 6). This effect has not been described in the literature before and is somewhat counterintuitive: our primary hypothesis was that LPI, being a proliferation stimulator, would decrease DHA-DA cytotoxicity. The cause of the observed action could be the fact that the GPR55-mediated DHA-DA cytotoxicity is realized via the intracellular Ca^2+^ concentration increase. GPR55 activation by LPI stimulates Ca^2+^ release as well [60], and thus could further increase this effect.

The obtained data clearly indicate that NADA are able to activate GPR55 and the downstream signaling typical for this receptor, and their activity was observed in the low micromolar range, similar to LPI [60]. Therefore, it could be speculated that these compounds could be viewed as a new class of GPR55 ligands. However, more detailed studies of the NADA affinity are required to prove this, but they were out of the scope of this work. The observed cytotoxicity mechanism of NADA may be specific only to those cell types capable of nNOS expression, and in other cell types cytotoxicity could be realized in a way similar to the one described for AA-EA [7]. In addition to that, NADA are known to activate both CB1 and TRPV1 receptors [13,48], which are capable of cancer cell death induction [62]. Given the fact that GPR55 blockers did not fully prevent NADA cytotoxicity in this study, these two receptors could be hypothesized to participate in this process as well.

The discovered ability of LPI to enhance NADA cytotoxicity is of particular interest. In the tumor setting, a large quantity of LPI is usually present and activates GPR55 with a consequent increase of cancer cell proliferation [59]. Besides, LPI enhanced serum-induced migratory and invasive responses in MDA-MB-231 cells, and this effect was mitigated by GPR55 siRNA [40]. The addition of NADA could thus switch this pro-proliferative activity into an antiproliferative one. The apparent switch of the LPI activity from pro-proliferative to pro-cytotoxic could be explained by the overactivation of the Ca^2+^ linked pathway of the GPR55 signaling during the concomitant action of DHA-DA and LPI, but this hypothesis requires further testing. As far as some synthetic agonists of GPR55 were recently described to induce cell death in breast cancer cells [37], it could be worthwhile testing whether the LPI cytotoxicity enhancement also takes place for other GPR55 ligands or is specific to the NADA class.

NADA and their representative DHA-DA are endogenous molecules; the possibility of their biosynthesis was shown in liver and nervous tissues homogenates [63,64], and thus they may be a component of the organism’s anticancer defense mechanism. The detected quantities of NADA in animal tissues are, however, in the nanomolar range [65,66], and thus to validate this hypothesis, a more detailed understanding on the regulation of the biosynthesis of these molecules is required.

## 4. Materials and Methods

### 4.1. Reagents

Isopropanol, MTT, D-glucose, DMSO, RPMI 1640, DMEM, L-glutamine, HCl, Triton X-100, Hank’s salts solution, Versene’s solution, penicillin, streptomycin, amphotericin B, RPMI 1640, DMEM, trypsin, (4,5-dimethylthiazol-2-yl)-2,5-diphenyltetrazolium bromide (MTT), and fetal bovine serum were from PanEco, Moscow, Russia.

Cell lines were purchased from ATCC, Manassas, VA, USA.

Antibodies anti-b-actin and anti-GPR55 were from Abcam, Cambridge, UK. Anti-mouse IgG antibody was from Santa-Cruz Biotechnology, Dallas, TX, US.

666-15, NQDI1, SP 600125, SB 202190, luciferin, and CoA were from Cayman Europe, Hamburg, Germany. O-1819, ARL 1477, CID 16020046, and U-73122 were from Tocris Bioscience, Bristol, UK. Glycylglycine, acetic acid, MgSO_4,_ EGTA, dithiothreitol, ATP, Pluronic F-62, DMSO, anti-nNOS antibody, Triton X-100, PTIO, *N*-Ac-Cys, cannabidiol, L-NAME, L-NMMA, acrylamide, bis-acrylamide, SDS, nitro blue tetrazolium, Tris-Borate-EDTA, agarose, bicinchoninic acid, D-glucose, bovine serum albumin, anti-rabbit IgG antibody, 5-Bromo-4-chloro-3-indolyl phosphate-toluoidine, Pluronic F-62, Fluorometric Intracellular Ros Kit, and Griess reaction components were from Sigma-Aldrich, St. Louis, MO, USA. Calcium Green, siRNA, RNAiMax, Advanced DMEM, and DreamTaq master mix were from Thermo Fisher Scientific, Walthon, MA USA. pCREB-Luc CREB luciferase reporter vector was from Signosis, Santa Clara, CA, USA. Total RNA Purification kit was from Jena Biosciences, Jena, Germany. MMLV reverse transcription kit, Red fluorescent protein expression vector pTagRFP-N were from Evrogen, Moscow, Russia. FuGENE HD transfection reagent was from Promega, Madison, WI, USA. The purity of all used reagents was 95% or more.

### 4.2. Chemical Synthesis

Amides of arachidonic acid with ethanolamine [67], cyclopropylamine [68] and 2-chloroethylamine (ethanolamine analog in which the hydroxyl group is replaced by chlorine atom) [69], dopamine amides of arachidonic, oleic, and docosahexaenoic acids [69], and arachidonoyl serotonin [70] were synthesized as described previously.

Virodamine was synthesized as described in [71] with small modification that concerned the method of activation of fatty acid carboxyl group. To a solution of arachidonic acid (125 mg, 0.4 mmole) in 3 mL acetonitrile at 0–4 °C 56 µL Py (1.7 equiv) and 58 µL CyF (1.7 equiv.) were added under stirring in argon atmosphere. After 90 min the resulting fluorine anhydride was extracted with hexane (15 mL). The mixture was evaporated to dryness, the residue was dissolved in 2 mL acetonitrile. The solution of 66 mg (0.4 mmole) BOC-protected ethanolamine in 0.5 mL MeCN and DMAP (50 mg, 0.4 mmole, 1.2 equiv) was added to fluorine anhydride under stirring in argon atmosphere. After 90 min the mixture was evaporated to dryness, diluted with water and extracted with ethyl acetate. The organic extracts were washed with water and drained, dried over sodium sulfate and concentrated under vacuum. Finally, the compound was isolated by column chromatography. The appropriate Boc-derivative was dissolved in TFA 5% in CH_2_Cl_2_ (3 mL) and stirred overnight [72]. The solvent was removed and crude was purified by column chromatography (CH_3_Cl/MeOH 95:5 containing 0.5% TFA) to give the desired compound. The final yield was 73%.

The structures of the synthesized compounds were confirmed using ESI-MS and ^1^H NMR. The purity of the compounds was confirmed using the reverse phase HPLC on a C18 column (Prontosil 120-3-C18 AQ, 2.0 × 75 mm, 4 µm particle size, EcoNova, Novosibirsk, Russia) in the acetonitrile: H_2_O gradient and was in the range of 97 to 99%.

### 4.3. Cell Culture

All cell lines were maintained in a CO_2_ incubator at 37 °C, 95% humidity and 5% CO_2_. The composition of the culture medium for the cells was as follows: PC12 (ATCC CRL-1721): RPMI 1640, 2 mM L-glutamine (L-Gln), and 7.5% fetal bovine serum (FBS); MCF-10A (ATCC CRL-10317), MDA-MB-231 (ATCC HTB-26), PANC-1 (ATCC CRL-1469), and SW620 (ATCC CCL-227): DMEM, 4 mM L-Gln, and 10% FBS, U-87 MG (ATCC HTB-14): MEM, 2 mM L-Gln, 1% nonessential amino acids, 1 mM pyruvate, and 10% FBS; DU 145 (ATCC HTB-81) and PC-3 (ATCC CRL-1435): RPMI 1640, 2 mM L-Gln, and 10% FBS. The cells were routinely checked for a mycoplasma contamination using RT-PCR. All cell media contained 100 U/mL penicillin, 100 µg/mL streptomycin, and 2.5 µg/mL amphotericin B. The cells were passaged using Trypsin-EDTA solution (PanEco, Moscow, Russia), the continuous passaging time did not exceed 40 passages.

### 4.4. Cytotoxicity, NO, and ROS Generation

For the analysis of cell death induction, NO production, and ROS generation, the cells were plated in 96-well plates at a density of 1.5 × 10^4^ cells per well and grown overnight. The dilutions of the test compounds prepared in DMSO and dissolved in the culture medium (without serum starvation) were added to the cells in triplicate for each concentration (100 µL of the fresh medium with the substance to 100 µL of the old medium in the well) and incubated for 18 h; the inhibitors were added 1 h before the substances (in that case, the medium with the substance also contained the inhibitor, and the added volumes were 50 µL for the inhibitor and 50 µL for the substance with the inhibitor). The incubation time was chosen based on the most pronounced differences between the compounds tested. The final DMSO concentration was 0.5%. Negative control cells (100% viability) were treated with 0.5% DMSO. Positive control cells (100% cell death) were treated with 3.6 μL of 50% Triton X-100 in ethanol per 200 μL of the cell culture medium. Separate controls were without DMSO (no difference with the control 0.5% DMSO was found). Depending on the experiment series, the effect of the test substances on the cell viability, NO and ROS production was evaluated using the MTT assay, Griess reaction, and Fluorometric Intracellular Ros Kit (Sigma-Aldrich, St. Louis, MO, USA), accordingly.

### 4.5. MTT Assay

For the MTT assay [73], after the removal of the medium with the test compounds, the cells and the controls were incubated for 1.5 h with 0.5 mg/mL of MTT in Hank’s salts solution, supplemented with 10 mM of D-glucose. After this incubation, an equal volume of 0.1 N HCl in isopropanol was added to each well and incubated at 37 °C for 30 min with shaking. The amount of the reduced dye was determined colorimetrically at 594 nm with a reference wavelength 620 nm using an EFOS 9505 photometer (Sapphire, Moscow, Russia). Additionally, the attachment and cell shape of the cells were evaluated microscopically.

### 4.6. ROS Assay

Reactive oxygen species accumulation was measured using the Fluorometric Intracellular Ros Kit (Sigma-Aldrich, St. Louis, MO, USA) according to the manufacturer’s protocol. In brief, after the incubation with the substances, cell medium was removed, and 100 µL of the ROS Detection Master Mix was added to each well and incubated for 30 min in CO_2_ incubator at 37 °C. After the incubation, the fluorescence intensity was determined using the microplate reader (Hidex Sense Beta Plus, Hidex, Turku, Finland) at λ_ex_ = 620 nm, λ_em_ = 665 nm.

### 4.7. NO Production Assay

NO generation was measured indirectly colorimetrically using the Griess reaction to the NO degradation product NO_2_^−^, as described earlier [74]. In brief, to 50 µL of the incubation medium, 50 µL of the 1:1 mixture of 2% sulfanilamide in 5% H_3_PO_4_ and 0.04% *N*-1-naphtyletylenediamine in H_2_O was added and incubated for 10 min at room temperature in darkness. After that, the optical density of the reaction mixture was determined photometrically at λ_max_ = 540 nm using EFOS 9505 plate reader (Sapphire, Moscow, Russia). The concentration curve was generated using a freshly prepared NaNO_2_ solution in PBS with PBS as the blank control.

### 4.8. GPR55 Knockdown

The GPR55 receptor knockdown was performed using a combination of three siRNAs [75] (siRNA 1: direct 5′-GGAGACAGCUGGAAUACAUTT-3′, reverse 5′-AUGUAUUCCAGCUGUCUCCTT-3′, siRNA 2: direct 5′-CGAAAGAGAGCCUGCAUCATT-3′, reverse 5′-UGAUGCAGGCUCUCUUUCGTT-3′, siRNA 3: direct 5′-GCAGAGUGAAGCAGGGCAUTT-3′, reverse 5′-AUGCCCUGCUUCACUCUGCTT-3′). Control cells were transfected with the scrambled siRNA (Thermo Fisher Scientific, Walthon, MA USA). The transfection was performed using the RNAiMax reagent (Thermo Fisher Scientific, Walthon, MA USA) according to the manufacturer’s protocol. Cells were seeded in a 96-well plate at the density of 30,000 cells in 100 µL of the appropriate full culture medium per well the day before the transfection. On the day of the transfection, 1 pmol of siRNA was diluted in 5 µL of the serum-free Advanced DMEM (Thermo Fisher Scientific, Walthon, MA USA), then mixed with 0.3 µL of the transfection reagent in 5 µL of the same medium 10 µL of the medium, incubated at room temperature for 5 min and added to the cells without medium replacement; 0.13 pmol of each siRNA was added per well. The knockdown efficiency was evaluated after 24, 48 and 72 h using RT-PCR. The substance cytotoxicity for the cells with GPR55 knockdown was evaluated 72 h after transfection. The cells were cultivated till the RT-PCR or substance treatment without medium change.

### 4.9. RNA Isolation and RT-PCR

Total RNA was isolated using the Total RNA Purification Kit (Jena Biosciences) according to the manufacturer’s protocol. Residual genomic DNA was removed using DNase I (Thermo Fisher Scientific, Walthon, MA USA) according to the manufacturer’s protocol; 1 U of the enzyme was used per RNA sample. cDNA was synthesized using the MMLV reverse transcription kit (Evrogen, Moscow, Russia) with an oligo-dT primer. PCR was performed using the DreamTaq master mix (Thermo Fisher Scientific, Walthon, MA USA); the program was as follows. Initial denaturation at 95 °C for 3 min, cycle: denaturation at 95 °C for 30 s, annealing at 57 °C for 30 s, DNA synthesis at 72 °C for 30 s for 35 cycles, final DNA synthesis at 72 °C for 5 min. The primers were generated using the ITDNA PrimerQuest tool (https://eu.idtdna.com/PrimerQuest; accessed on 01.01.2020) and validated using the NCBI Primer-BLAST service [76]. Their sequences were as follows: GPR55 forward 5′-CTCCCTCCCATTCAAGATGA-3′, reverse 5′-AAGATCTCCAGGGGGAAGAA-3′, GAPDH forward 5′-GGGCGCCTGGTCACCA-3′, reverse 5′-AACATGGGGGCATCAGCAGA-3′; iNOS forward 5′-CAAACCTTCCGGGCAGCCTGTGAGACG-3′, reverse 5′-AAAATCCCCAGGTGTTCCCCAGGTAGGTAGC-3′; eNOS forward 5′-AAGTGGGCAGCATCACCTAC-3′, reverse 5′-CCGGGTGTCTAGATCCATGC-3′; nNOS forward 5′-GAAGGCGAACAACTCCCTCA-3′, reverse 5′-CTGATTCCCGTTGGTGTGGA-3′. Reaction mixtures without reverse transcriptase and without cDNA matrix were used as negative controls.

### 4.10. Western Blot

To evaluate the expression of particular proteins in the cells, the cells were seeded at the density 200,000 per well of 24-well plate the day before experiment. After the appropriate treatment, the cells were washed once with PBS, lysed using the lysis solution (150 mM NaCl, 1% Triton X-100, 0.1% SDS, 50 mM Tris-HCl pH 8.0, 1% protease inhibitor cocktail) for 30 min at +4 °C, and centrifuged for 5 min at 10,000× *g*. Total protein concentration in the supernatants was determined using the BCA assay. Proteins were separated using denaturing SDS-PAGE in 10% gel with the PageRuler protein ladder, transferred to a nitrocellulose membrane using the Invitrogen Power Blotter with the Invitrogen Power Blotter 1-step transfer buffer and Invitrogen precut membranes and filters, and stained with antibodies using the Invitrogen iBind system according to the manufacturer’s protocol. The following antibodies were used: rabbit anti-GPR55 (Abcam ab203663), mouse anti-beta-actin (Abcam ab8226), mouse anti-nNOS (Sigma-Aldrich N2280, St. Louis, MO, USA); secondary antibodies (coupled to alkaline phosphatase) anti-rabbit IgG (Sigma-Aldrich A9919, St. Louis, MO, USA), anti-mouse IgG (Santa-Cruz Biotech scbt-2008). After the staining, the membrane was washed in H_2_O for 10 min and incubated with the staining solution (20 µL BCIP solution + 30 µL NBT solution per 10 mL of substrate buffer) for 1 h at room temperature. Substrate buffer for alkaline phosphatase: 100 mM Tris-HCl, pH 9.5, 100 mM NaCl, and 5 mM MgCl_2_. BCIP solution: 20 mg/mL 5-Bromo-4-chloro-3-indolyl phosphate-toluoidine (BCIP) in 100% di-methyl formamide. NBT staining solution: 50 mg/mL nitro blue tetrazolium (NBT) in 70% di-methyl formamide.

### 4.11. BCA Protein Assay

Protein concentration was determined using the BCA assay [77]. The following base reagents were used: Reagent A (bicinchoninic acid 1%, Na_2_CO_3_*H_2_O 2%, sodium tartrate 0.16%, NaOH 0.4%, NaHCO_3_ 0.95%, pH 11.25), Reagent B (4% CuSO_4_*5H_2_O), S-WR (50 volumes of Reagent A + 1 volume of Reagent B). 5 µL of cell lysate was mixed with 40 µL of S-WR and incubated for 15 min at 60 °C, after which the optical density was measured at λ = 562 nm using the Hidex Sense Beta Plus microplate reader (Hidex, Turku, Finland). Each sample was assayed in triplicate. Cell lysis buffer was used as a background control. Bovine serum albumin solution in the cell lysis buffer was used as a positive control and to build a calibration curve.

### 4.12. Calcium Measurements

To measure the calcium response to the DHA-DA treatment, the cells were plated in 96-well plates at the density 30,000 cells/cm^2^. For the experiments, 1 mg/mL D-glucose and 1 mg/mL fatty acid free bovine serum albumin in Hank’s salts were used as an incubation medium. Before treatment, the cells were loaded with 1.5 mM calcium green dye in the incubation medium containing 0.4% Pluronic F-62 for 1 h at 37 °C with culture medium removal but without prior washing. Then, the cells were washed three times with 200 µL of incubation medium and incubated with inhibitors or incubation medium alone for 1 h at 37 °C. Finally, an equal volume of incubation medium with NADA alone or in combination with the inhibitors already present in wells was added to each well. The measurement was carried out using the Hidex Sense Beta Plus microplate reader (Hidex, Turku, Finland) at 25 °C without shaking, λ_ex_ = 485 nm, λ_em_ = 535 nm.

### 4.13. CREB Activation Assay

CREB transcription factor activation was measured using the pCREB-Luc CREB luciferase reporter vector (Signosis, Santa Clara, CA, USA). The cells were seeded at the density of 60,000 per well of 96-well plate the day before transfection and were cotransfected with the vector and the red fluorescent protein expression vector pTagRFP-N (Evrogen, Moscow, Russia) at a 1:1 ratio using the FuGENE HD transfection reagent (Promega, Madison, WI, USA) according to the manufacturer’s protocol. The day after the transfection the cells were treated with the test substances for 3 h in a fresh medium, after which the RFP fluorescence was measured, the cells were lysed and used for the luciferase activity detection.

### 4.14. Luciferase Activity Detection

To measure the luciferase activity after CREB stimulation, cell culture medium was removed and the cells were lysed in 15 µL of the lysis buffer (25 mM glycylglycine titrated with acetic acid to pH 8, 1% Triton X-100, 5 mM MgSO_4_, 4 mM EGTA, and 4 mM dithiothreitol) per well of a 96-well plate for 10 min at room temperature. After that, 50 µL of firefly luciferase assay solution (5 mM glycylglycine titrated with acetic acid to pH 8, 5 mM MgSO_4_, 150 µM luciferin, 2 mM ATP, 40 µM CoA, 4 mM EGTA, and 4 mM DTT) was added to each well, incubated in the dark for 5 s, and the luminescence was recorded using the Hidex Sense Beta Plus plate reader for 20 s in the 550–570 nm range.

### 4.15. Molecular Docking

For the docking studies, an optimized GPR55 model in the open state was used [5]. Ligand structures (LPI, PubChem CID 42607497, DHA-DA PubChem CID 78108376, AA-EA PubChem CID 5281969) were obtained from the PubChem database (https://pubchem.ncbi.nlm.nih.gov/, accessed on 01.01.2020) and optimized using the OpenBabel 2.4.1 software (http://openbabel.org/, accessed on 10.01.2020) [78]. Molecular docking was performed using the AutoDock Vina v. 1.1.2 (http://vina.scripps.edu/, accessed on 10.01.2020) [79]. To detect possible alternative binding sites and compare the affinities of the ligands for them, the procedure described in [80] was used. As such, molecular docking was performed in two steps: first, we docked each molecule to the whole receptor as one large binding area to locate potential alternative binding sites. Then, the coordinates of the docking results were clustered and averaged to give the centers of the binding sites, and the docking was repeated to a narrow zone around the new binding sites. The first docking was performed with the grid center coordinates (11.93, −2.87, 16.51) and grid size 30 × 32 × 26 Å chosen to cover the whole extracellular receptor part and exhaustiveness 8. For each ligand, the docking was performed 100 times generating 10 conformations each time. The resulting coordinates were clustered using the AgglomerativeClustering algorithm from the scikit-learn package [81] (linkage = “ward”, n_clusters = 5), and the coordinates in each cluster were averaged to obtain their centers. The number of clusters was chosen using the DBSCAN algorithm from the same package. After the determination of the cluster centers, each ligand was docked into each cluster’s centroid with grid size 20 × 20 × 20 Å and exhaustiveness 256 5 times generating 10 conformations each time.

### 4.16. Statistical Procedures

Each experiment was repeated three times. Every experimental point was done in triplicate wells. Data analysis was performed using the GraphPad Prism software (www.graphpad.com) v. 8.4.3. Data are presented as mean ± standard error. Data were compared using the unpaired Student’s *t* test for pairwise comparison and ANOVA with the Tukey post-test for multiple comparison; *p* values of 0.05 or less were considered significant.

## 5. Conclusions

In this study, among the endocannabinoids and similar compounds we identified NADA as the most active GPR55 activator, being able to induce cancer cell death via this receptor activation. In PC12 cell line, GPR55 activation by NADA resulted in Ca^2+^ release, nNOS expression and activity induction, and oxidative stress. The cytotoxicity of NADA was enhanced in the presence of the natural GPR55 ligand LPI, which normally acts as a proliferation inducer.

## Figures and Tables

**Figure 1 ijms-22-00622-f001:**
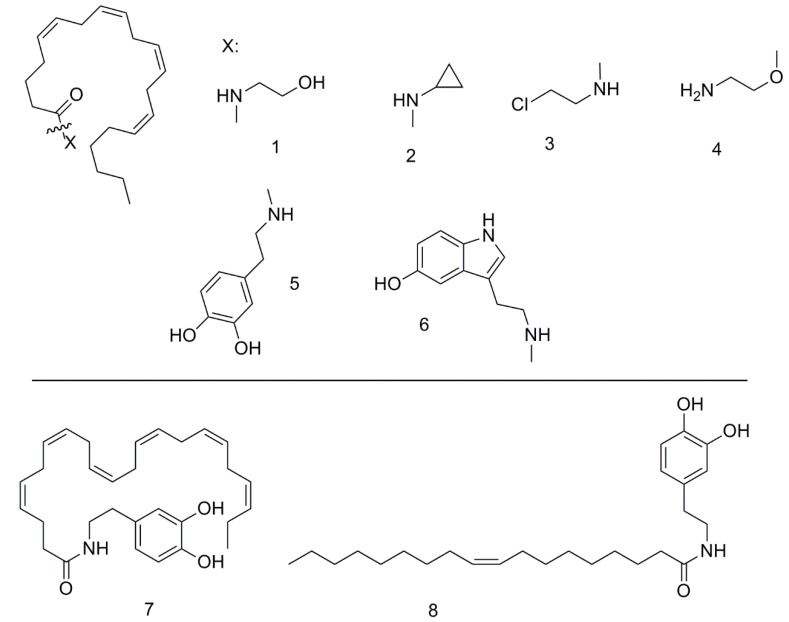
The structures of the compounds used in this work: 1, *N*-arachidonyl ethanolamine (AAEA), 2, arachidonoyl cyclopropylamine, 3, arachidonoyl 2-chlorethylamine, 4, O-arachidonoyl ethanolamine (VRD), 5, arachidonoyl dopamine, 6, arachidonoyl serotonin, 7, docosahexaenoyl dopamine, 8, oleoyl dopamine.

**Figure 2 ijms-22-00622-f002:**
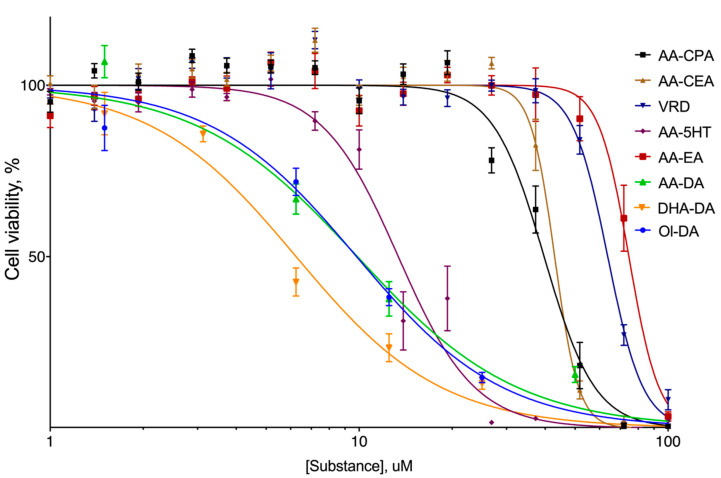
The cytotoxicity of the synthesized fatty acid esters and ethers for the undifferentiated PC12 cell line, 24 h incubation time, MTT assay. *n* = 5 experiments, M. ± S.E. AA-CPA, *N*-arachidonoyl cyclopropyamine, AA-CEA, *N*-arachidonoyl 2-chloroethylamine, VRD, virodhamine, AA-5HT, *N*-arachidonoyl serotonin, AA-EA, anandamide, AA-DA, *N*-arachidonoyl dopamine, DHA-DA, *N*-docosahexaenoyl dopamine, Ol-DA, N-oleoyl dopamine.

**Figure 3 ijms-22-00622-f003:**
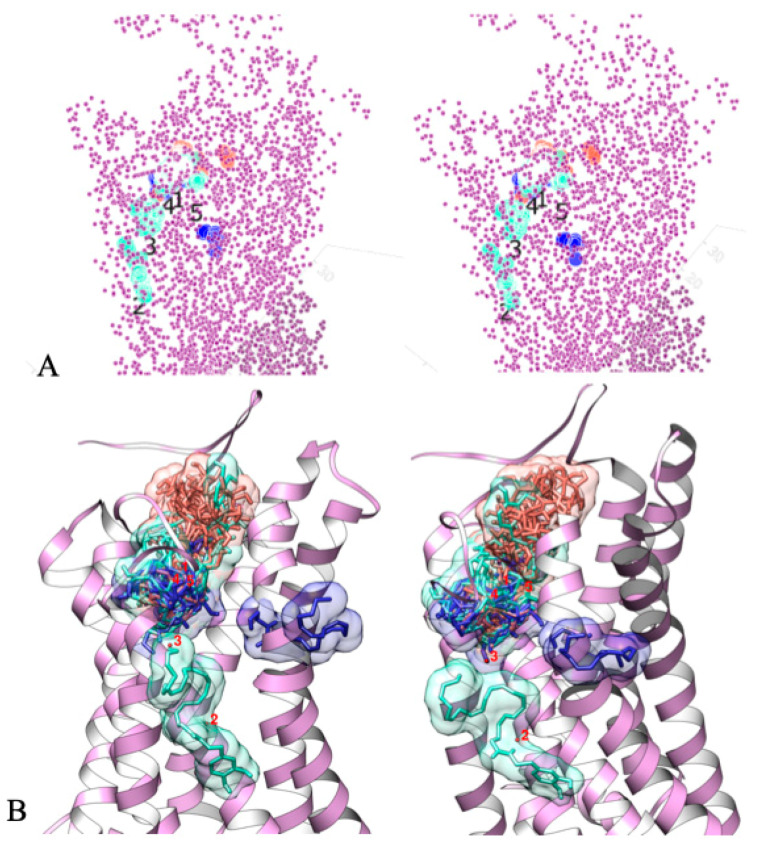
Alternative binding sites of anandamide (AA-EA, blue), lysophosphatydyl inositol (LPI, red), and *N*-docosahexaenoyl dopamine (DHA-DA, cyan) on GPR55 (shown in purple). Molecular docking results (AutoDock Vina); (**A**) docking result centers and GPR55 atom coordinates, (**B**) some docked ligands on GPR55 ribbon model (same coloring) and cluster numbering.

**Figure 4 ijms-22-00622-f004:**
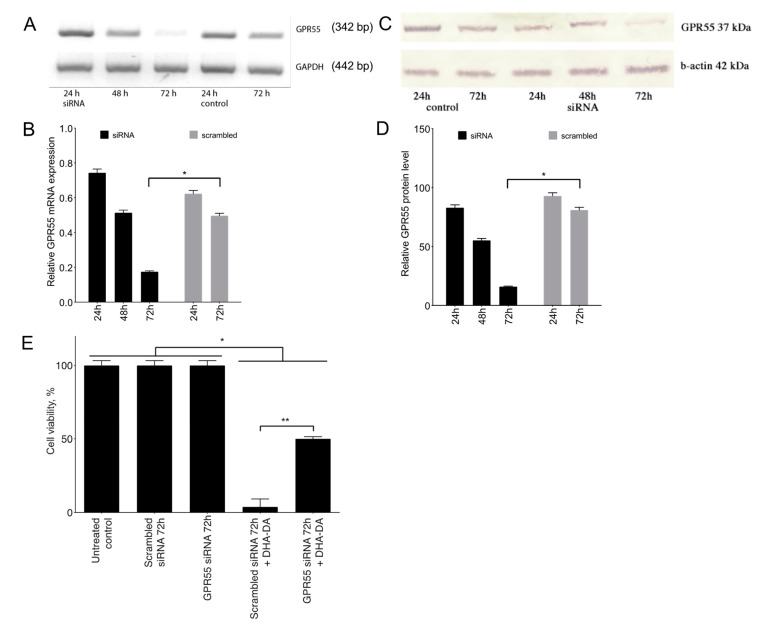
GPR55 knockdown influence on *N*-docosahexaenoyl dopamine (DHA-DA) cytotoxicity for PC12 cells. (**A**), RT-PCR of GPR55 mRNA after siRNA treatment. (**B**), RT-PCR quantification, densitometry data and normalized to the GAPDH housekeeping gene. (**C**), Western blot of GPR55 after siRNA treatment. (**D**), Western blot quantification, densitometry data and normalized to b-actin. Data are presented as mean ± S.E., *n* = 3 experiments. *, a statistically significant difference from the appropriate control treated with a scrambled siRNA, ANOVA with the Tukey post-test, *p* ≤ 0.05. (**E**), DHA-DA (16 μM) toxicity for the PC12 cells after 72 h of the transfection with either the GPR55 siRNA or the scrambled control RNA; 24 h incubation time, MTT assay, mean ± S.E., *n* = 3 experiments. *, a statistically significant difference from the DMSO control, ANOVA with the Tukey post-test, *p* ≤ 0.05; **, a statistically significant difference between DHA-DA toxicity for the cells treated with GPR55 siRNA and scrambled siRNA, unpaired Student’s *t*-test, *p* ≤ 0.05.

**Figure 5 ijms-22-00622-f005:**
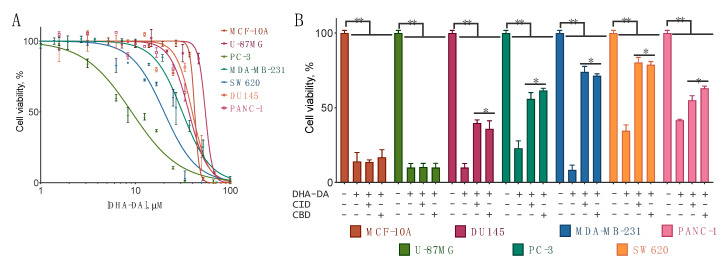
Cytotoxicity of *N*-docosahexaenoyl dopamine (DHA-DA) for the cell lines without (MCF-10A, U-87 MG) and with (all other lines) the GPR55 receptor. Incubation time 24 h, MTT assay, mean ± S.E., *n* = 3 experiments. (**A**) Dose-effect curves for the cell lines with GPR55. (**B**) The activity of DHA-DA (the concentration was chosen as EC_75_ or EC_50_, cf. Table 3) with the addition of the GPR55 blockers CID16020046 (CID, 0.4 μM) and cannabidiol (CBD, 0.4 µM). The blockers did not affect cell viability at these concentrations. *, a statistically significant difference from the control without inhibitor for the appropriate cell line, **, a statistically significant difference from the DMSO control, ANOVA with the Tukey post-test, *p* ≤ 0.05.

**Figure 6 ijms-22-00622-f006:**
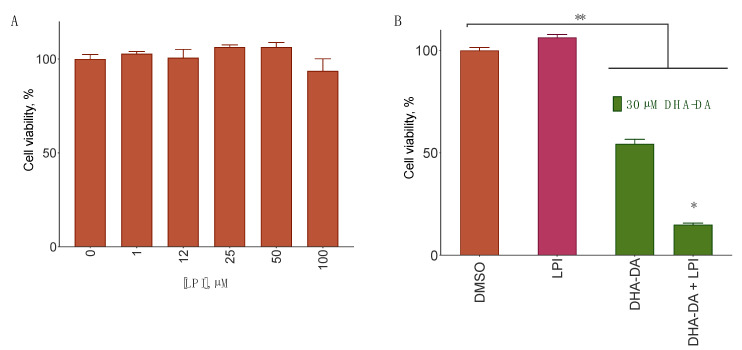
Lysophosphatidyl inositol (LPI) interplay with *N*-docosahexaenoyl dopamine (DHA-DA) on MDA-MB-231 cells in the cell viability model. (**A**), the influence of LPI on MDA-MB-231 cells viability. (**B**), the influence of LPI (50 µM) on DHA-DA cytotoxicity for MDA-MB-231 cells. 24 h incubation, MTT assay, mean ± S.E., *n* = 3 experiments. *, a statistically significant difference from the appropriate concentration of DHA-DA alone, **, a statistically significant difference from the DMSO control, ANOVA with the Tukey post-test, *p* ≤ 0.05.

**Figure 7 ijms-22-00622-f007:**
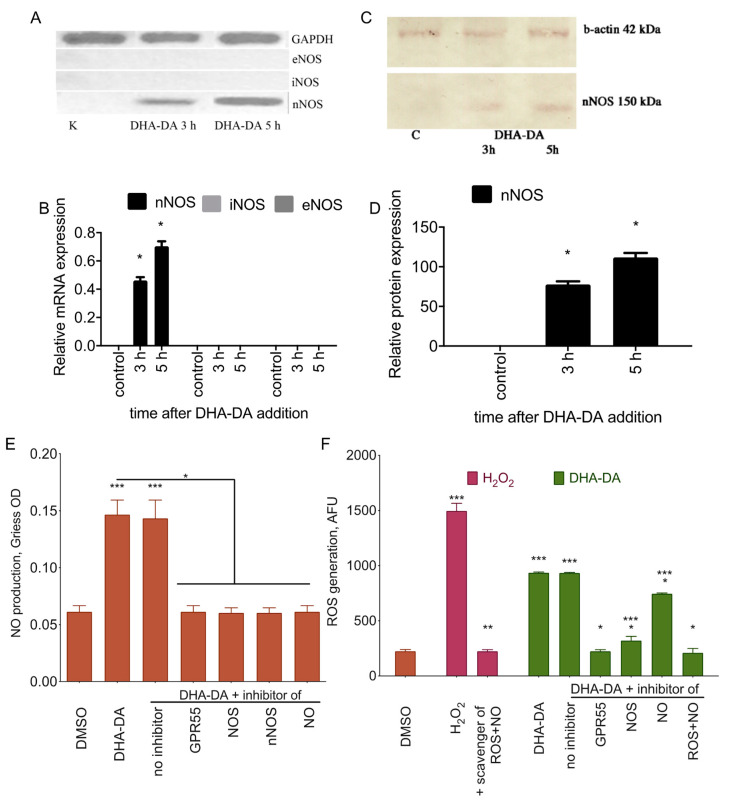
*N*-docosahexaenoyl dopamine (DHA-DA) induces neuronal NO synthase (nNOS) mRNA expression with a consequent ROS and NO production. (**A**,**B**) NO synthase mRNA expression (iNOS, inducible NO synthase, eNOS, endothelial NO synthase), (**C**,**D**) NO synthase protein expression, (**E**) NO and (**F**) ROS production after the undifferentiated PC12 treatment with DHA-DA (16 μM) in comparison with hydrogen peroxide (500 μM) ROS induction. Incubation time for E and F was 42 h. Blockers and inhibitors: GPR55, O-1918 (3 μM), N OS, L-NMMA (3 μM), nNOS, ARL 1477 (10 μM), NO, PTIO (5 μM), ROS + NO, Ac-Cys (5 µM). NO was detected using the Griess assay, and ROS was detected using a fluorometric ROS detection kit. Data are presented as mean ± S.E., *n* = 3 experiments. *, a statistically significant difference from the DHA-DA without inhibitors; **, a statistically significant difference from H_2_O_2_ without inhibitors; ***, a statistically significant difference from DMSO, ANOVA with the Tukey post-test, *p* ≤ 0.05. NOS isoforms mRNA expression was detected using RT-PCR, quantified using densitometry and normalized to the GAPDH housekeeping gene; data are presented as mean ± S.E., *n* = 3 experiments. nNOS protein expression was detected using Western blotting with antibody staining, quantified using densitometry and normalized to the b-actin; data are presented as mean ± S.E., *n* = 3 experiments. *, a statistically significant difference from untreated control, ANOVA with the Tukey post-test, *p* ≤ 0.05.

**Figure 8 ijms-22-00622-f008:**
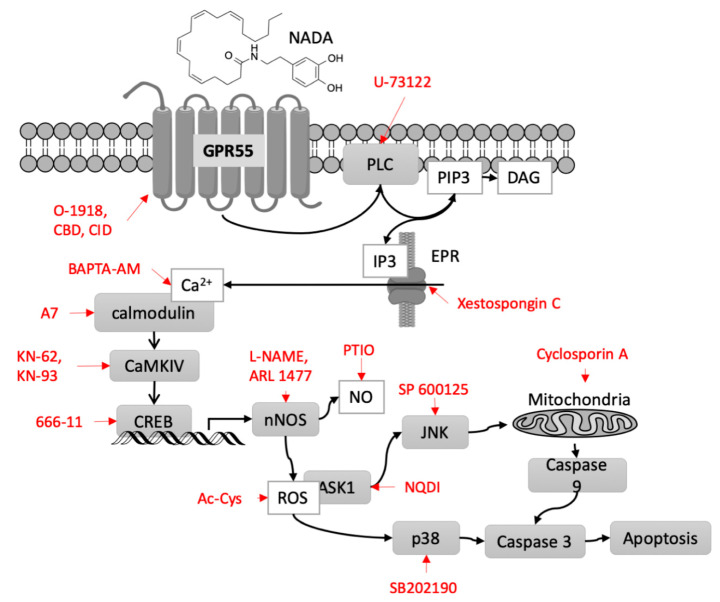
Proposed signal transduction during the cell death induction by endovanilloid DHA-DA. Inhibitors used in the work are shown in red. nNOS, neuronal NO synthase, JNK, c-Jun N-terminal kinase, ASK1, Apoptosis signal-regulating kinase 1, CBD, cannabidiol, CREB, cAMP response element-binding protein, EPR, endoplasmic reticulum, PIP3, Phosphatidylinositol (3,4,5)-trisphosphate, DAG, diacylglycerol, PLC, phospholipase C, IP3, Inositol trisphosphate, CaMKIV, Calcium/calmodulin-dependent protein kinase type IV, NADA, *N*-acyl dopamines.

**Figure 9 ijms-22-00622-f009:**
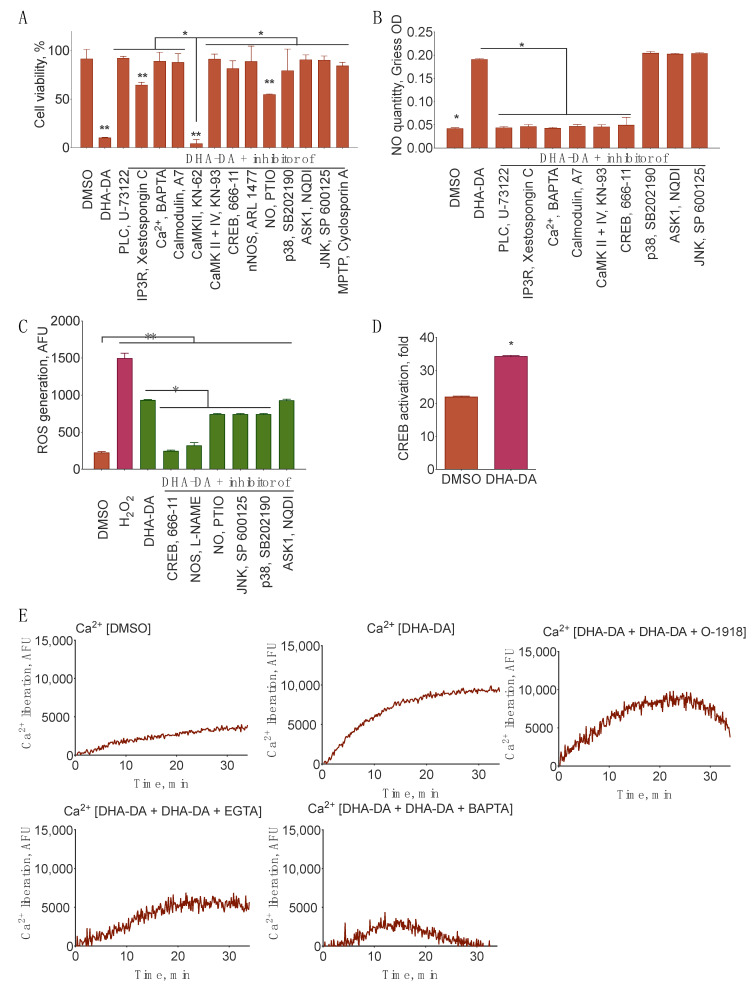
Signal transduction from GPR55 to its downstream targets during cell death induction in undifferentiated PC12 cells. (**A**) *N*-docosahexaenoyl dopamine (DHA-DA, 16 µM) cytotoxicity (MTT assay), (**B**) NO accumulation induction (Griess assay), and (**C**) ROS induction (ROS detection kit) with the inhibitors of phospholipase C (PLC, U-73122, 3 μM), inositol triphosphate receptor (IP3R, Xestospongin C, 10 μM), Ca^2+^ chelator (BAPTA-AM, 10 μM), Calmodulin (A7, 10 µM), CaMKII (KN-62, 10 µM), calcium/calmodulin activated kinase type II and IV (CaMKII + IV, KN-93), cAMP response element binding protein (CREB, 666-11, 5 μM), neuronal NO synthase (nNOS, ARL 1477, 10 µM), NO scavenger (PTIO, 10 µM), p38 (SB202190, 15 µM), apoptosis signal-regulated kinase 1 (ASK1, NQDI, 1 µM), c-Jun N-terminal kinase (JNK, SP 600125, 5 µM), Mitochondrial permeability transition pore MPTP (Cyclosporin A, 10 μM), NOS (L-NAME, 10 µM), and NO chelator (PTIO, 10 µM). (**D**) CREB (cAMP response element binding protein) activation by DHA-DA (16 µM), cells transfected with a CREB luciferase reporter and an RFP plasmid, Ratio Firefly luminescence/RFP fluorescence. (**E**) Ca^2+^ response of the cells preloaded with calcium green dye after the treatment with DHA-DA (16 µM) in combination with either of the extracellular (EGTA, 5 mM) and intracellular (BAPTA-AM, 50 µM) Ca^2+^ chelators or GPR55 blocker (O-1918, 5 µM). Fluorescence of the Calcium green dye, Mean ± S.E., *n* = 3 experiments. *, statistically significant difference from control, **, statistically significant difference from DMSO (panels A and C), ANOVA with Tukey post-test, *p* ≤ 0.05.

**Table 1 ijms-22-00622-t001:** The cytotoxicity of the synthesized fatty acid esters and ethers for the undifferentiated PC12 cells, 24 h incubation time, MTT assay, EC_50_ values, lower value means higher cytotoxicity, mean with 95% confidence intervals, *n* = 5 experiments. AA-CPA, *N*-arachidonoyl cyclopropyamine, AA-CEA, *N*-arachidonoyl 2-chloroethylamine, VRD, virodhamine, AA-5HT, *N*-arachidonoyl serotonin, AA-EA, anandamide, AA-DA, *N*-arachidonoyl dopamine, DHA-DA, *N*-docosahexaenoyl dopamine, Ol-DA, N-oleoyl dopamine.

	AA-CEA	AA-CPA	VRD	AA-EA	AA-5HT	DHA-DA	Ol-DA	AA-DA
	μM, M. (95% Confidence Interval)							
EC_50_	36 (37.93 to 41.50)	40 (41.49 to 44.42)	67 (61.88 to 65.72)	80 (71.14 to 78.15)	14 (12.53 to 14.33)	6 (5.434 to 7.224)	10 (8.635 to 11.05)	10 (8.035 to 12.46)

**Table 2 ijms-22-00622-t002:** Calculated binding energies of anandamide (AA-EA), *N*-docosahexaenoyl dopamine (DHA-DA), and lysophosphatydyl inositol (LPI) for the calculated alternate DHA-DA binding clusters of GPR55 molecule, molecular docking results (AutoDock Vina); the lower energy corresponds to a higher affinity, mean ± S.E., *n* = 5 experiments.

DHA-DA		AA-EA		LPI	
Cluster	Dock Score, kcal/M	Cluster	Dock Score, kcal/M	Cluster	Dock Score, kcal/M
	Mean ± S.E.		Mean ± S.E.		Mean ± S.E.
3	−8.77 ± 0.03	1	−6.61 ± 0.02	3	−7.76 ± 0.04
1	−7.56 ± 0.02	3	−6.6 ± 0.03	1	−7.02 ± 0.03
5	−7.37 ± 0.02	4	−6.51 ± 0.02	4	−6.88 ± 0.03
4	−7.33 ± 0.04	5	−6.36 ± 0.03	5	−6.82 ± 0.03
2	−7.26 ± 0.04	2	−5.8 ± 0.05	2	−6.29 ± 0.03

**Table 3 ijms-22-00622-t003:** *N*-docosahexaenoyl dopamine (DHA-DA) cytotoxicity for various cell lines correlates with the GPR55 blocker activity. Incubation time 24 h, MTT assay, EC_50_ values (lower value means higher toxicity, mean with 95% confidence interval) and presence of blocker activity.

Cell Line	DHA-DA Cytotoxicity EC_50_, μM M. (95% C.I.)	DHA-DA Toxicity Change with GPR55 Blockers, Change in % of Cell Viability M. ± S.E.
MCF-10A	42 (39.67 to 43.23)	0
U-87MG	56 (54.21 to 58.02)	0
DU145	38 (36.20 to 43.39)	+30 ± 2
PC-3	9 (8.075 to 10.23)	+32 ± 1
MDA-MB-231	30 (29.45 to 33.11)	+70 ± 2
SW620	22 (17.79 to 22.59)	+41 ± 1
PANC-1	36 (34.90 to 38.51)	+22 ± 1

## Data Availability

The data presented in this study are available on request from the corresponding author. The data are not publicly available due to legal issues.

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
