# Peer review of "GPR55 Receptor Activation by the N-Acyl Dopamine Family Lipids Induces Apoptosis in Cancer Cells via the Nitric Oxide Synthase (nNOS) Over-Stimulation"

_ijms, 2021, doi:10.3390/ijms22020622_

Round 1
Reviewer 1 Report
It is preferable to demonstrate (by western blotting, with an appropriate positive control) that no NOS isoform (e.g., eNOS and iNOS) other than nNOS were present in the cancer cell line used in this study.
Otherwise, the authors have successfully addressed the concerns identified.
Author Response
Point 1. It is preferable to demonstrate (by western blotting, with an appropriate positive control) that no NOS isoform (e.g., eNOS and iNOS) other than nNOS were present in the cancer cell line used in this study.
Response 1. In this work, we used non-differentiated PC12 cells as a model. According to the literature data, neither eNOS, nor iNOS are expressed in such cells (Brain Res Mol Brain Res. 1997 Dec 1;52(1):71-7; Purinergic Signal. 2005 Jun;1(2):161-72); they only appear after a treatment with the differentiation inducting stimuli (J Nanobiotechnology. 2013 Oct 11;11:35). Therefore, given that we did not detect mRNA of these isoforms, an additional check by western blotting was not necessary. Moreover, the positive control for iNOS under our experimental conditions is impossible, because DHA-DA induced apoptosis rather than differentiation within the substance concentration used. The corresponding explanation with additional references was added to the manuscript.
Reviewer 2 Report
The authors replied adequately to major concern #1 and included appropriate references in the revised version of the manuscript.
On the other hand, the authors did not reply adequately to major concern #2. Authors keep on considering MCF-10A as cancer cells. Instead, these cells should be used as normal epithelial cells. It is crucial that DHA-DA is used in all the bioassays in a concentration that is not toxic in MCF-10A.
The authors corrected mistyings and addressed minor concerns to the first version.
Author Response
Point 1. Authors keep on considering MCF-10A as cancer cells. Instead, these cells should be used as normal epithelial cells. It is crucial that DHA-DA is used in all the bioassays in a concentration that is not toxic in MCF-10A.
Response 1. The text was changed to indicate that MCF-10A is a non-tumorigenic epithelial cell line (according to the ATCC description). In most experiments, except for the screening stage, the concentration of DHA-DA (<40 µM) used was non-toxic for the MCF-10 cell line. The data on the activity of high concentrations of DHA-DA in the combination with LPI was removed from the text.
Round 2
Reviewer 2 Report
The authors addressed my main concern regarding the use of toxic concentrations of DHA-DA and modified the manuscript accordingly. Therefore, the revised version of the manuscript is acceptable for publication.
This manuscript is a resubmission of an earlier submission. The following is a list of the peer review reports and author responses from that submission.
Round 1
Reviewer 1 Report
Manuscript ID: ijms-1009398
Title: GPR55 receptor activation by the N-acyl dopamine family lipids induces apoptosis in cancer cells via the nitric oxide synthase (nNOS) over-stimulation
General comment
In this manuscript, the authors chemically synthesized and tested a set of fatty acid amides and esters for the cell death induction via GPR55 activation. The most active compounds appeared to be N-acyl dopamines, especially N-docosahexaenoyl dopamine (DHA-DA). Moreover, these compounds were shown to induce cell death in a panel of cancer cell lines via a mechanism different from that published for anandamide and including neuronal nitric oxide synthase overstimulation with concomitant oxidative stress induction.
I have some concerns on the interpretation of the results and the conclusions. Moreover, the manuscript needs a deep revision in the introduction and in the discussion, according to the major concerns listed below.
Major
1) Since the authors claimed that the best results are obtained with DHA-DA, it is not adequate that they completely missed a large body of literature related to omega-3 fatty acid amides and their effects. Authors should consider the following reviews and articles.
Reviews:
Prostaglandins Lipid Mediat 2019, 144, 106351;
Prostaglandins Lipid Mediat 2019, 143, 106337;
Mol Aspects Med 2018, 64, 34;
Eur J Pharmacol 2016, 785, 96;
Br J Pharmacol 2013, 169, 772
Articles reporting the in vitro antitumor activities of DHA-DA in breast cancer cell lines:
Biochim Biophys Acta 2015, 1850, 2185
Articles reporting the in vitro antitumor activities of DHA-EA in prostate and breast cancer cell lines:
Carcinogenesis 2010, 31, 1584;
J Cell Physiol 2013, 228, 1314
Articles reporting the anti-inflammatory properties of DHA-DA, DHA-EA and DHA-serotonin, in particular the effects on NO and iNOS:
ACS Chemical Neuroscience, 2017, 8, 548;
Biochim Biophys Acta 2016, 1861, 2020;
Br J Pharmacol 2015, 172, 24;
Br J Nutr 2011, 105, 1798
2) The MCF-10A human breast epithelial cell line is the most commonly used normal breast cell model (PlosONE 2015, 10(7), e0131285). It is crucial that any treatment on cancer cells induce limited (or none) damage on normal cells. Therefore, each compound should be tested in the cancer cell lines in a concentration range that evoke no cytotoxic effects in MCF-10A.
Minor
1) In several parts of the manuscript citations (?) are not correctly displayed and the text “(Error! Reference source not found.)” appears.
2) “N” should be in italic in “N-acyl”, “N-arachidonoyl” and so on.
3) In figure 1 and throughout the text, please change “arachidonyl ethanolamide” with “arachidonoyl ethanolamine”.
Reviewer 2 Report
This is an interesting study examining the role of GPR55 modulation in inducing cancer cell death and investigating the potential pathways downstream of GPR55. However, there are several concerns with this manuscript as described below:
- The key findings using mRNA expression changes must be furtehr validated by corresponding protein levels (e.g., by western blotting). Since the presence of the actual expressed protein is needed for a biological event, these data must be shown. This is especially important for data pertaining to GPR55 levels (in context of siRNA studies; fig 4) and nNOS levels (following addition of cell death inducers; fig 7).
- In Figure 5/ Table 3, the actual protein level of nNOS and GPR55 in each of the cancer cell lines must be demonstrated.
- In figure 7, presence of eNOS and iNOS protein must be examined since an absence of mRNA expression does not preclude basal levels of protein expressions.
- It is quite concerning that many of the cells in various experiments were exposed to high (0.5%) levels of DMSO. DMSO can have an antioxidant effect on its own so teh results of DHA-DA+ DMSO must be shown on both NO and ROS generation
- More details on the type/nature of ROS generated is needed. Does the assay kit detect superoxide alone, or both superoxide and hydrogen peroxide. The possibility of NO reacting with superoxide and generating peroxynitrite must be examined. There are specific ROS-detecting assays and fluorescent dyes which can be used to determine the exact nature of ROS generated (and its interactions, if any, with NO). If some of the experiments cannot be performed, the proper justification should be provided and the possibilities discussed in detail.
- The descriptors in the Y-axis of graphs should be presented vertically, instead of in a slanted manner, to prevent any confusion and/or misunderstanding.